# Immunohistochemical Demonstration of the pGlu79 α-Synuclein Fragment in Alzheimer’s Disease and Its Tg2576 Mouse Model

**DOI:** 10.3390/biom12071006

**Published:** 2022-07-20

**Authors:** Alexandra Bluhm, Sarah Schrempel, Stephan Schilling, Stephan von Hörsten, Anja Schulze, Steffen Roßner, Maike Hartlage-Rübsamen

**Affiliations:** 1Paul Flechsig Institute for Brain Research, University of Leipzig, 04103 Leipzig, Germany; alexandra.bluhm@medizin.uni-leipzig.de (A.B.); sarah.schrempel@medizin.uni-leipzig.de (Sa.S.); maikerbs@rz.uni-leipzig.de (M.H.-R.); 2Fraunhofer Institute for Cell Therapy and Immunology, Department of Drug Design and Target Validation, 06120 Halle (Saale), Germany; stephan.schilling@izi.fraunhofer.de (S.S.); anja.schulze@izi.fraunhofer.de (A.S.); 3Faculty of Applied Biosciences and Process Engineering, Anhalt University of Applied Sciences, 06366 Köthen, Germany; 4Department for Experimental Therapy, University Clinics Erlangen and Preclinical Experimental Center, University of Erlangen-Nuremberg, 91054 Erlangen, Germany; stephan.v.hoersten@fau.de

**Keywords:** Alzheimer’s disease, β-amyloid, α-synuclein, matrix metalloproteinase-3, glutaminyl cyclase, heat shock protein 27, reactive astrocytes

## Abstract

The deposition of β-amyloid peptides and of α-synuclein proteins is a neuropathological hallmark in the brains of Alzheimer’s disease (AD) and Parkinson’s disease (PD) subjects, respectively. However, there is accumulative evidence that both proteins are not exclusive for their clinical entity but instead co-exist and interact with each other. Here, we investigated the presence of a newly identified, pyroglutamate79-modified α-synuclein variant (pGlu79-aSyn)—along with the enzyme matrix metalloproteinase-3 (MMP-3) and glutaminyl cyclase (QC) implicated in its formation—in AD and in the transgenic Tg2576 AD mouse model. In the human brain, pGlu79-aSyn was detected in cortical pyramidal neurons, with more distinct labeling in AD compared to control brain tissue. Using immunohistochemical double and triple labelings and confocal laser scanning microscopy, we demonstrate an association of pGlu79-aSyn, MMP-3 and QC with β-amyloid plaques. In addition, pGlu79-aSyn and QC were present in amyloid plaque-associated reactive astrocytes that were also immunoreactive for the chaperone heat shock protein 27 (HSP27). Our data are consistent for the transgenic mouse model and the human clinical condition. We conclude that pGlu79-aSyn can be generated extracellularly or within reactive astrocytes, accumulates in proximity to β-amyloid plaques and induces an astrocytic protein unfolding mechanism involving HSP27.

## 1. Introduction

Alzheimer’s disease (AD) is the most prevalent neurodegenerative disorder affecting 55 million people worldwide. It is histopathologically characterized by intracellular neurofibrillary tangles composed of abnormally phosphorylated tau protein and by extracellular β-amyloid (Aβ) deposits [1,2,3]. Neurodegeneration in AD affects cortical, hippocampal and subcortical structures, including nucleus basalis Meynert [4,5], locus coeruleus [6,7,8] and Edinger Westphal nucleus [9,10,11].

Parkinson’s disease (PD) is the second most frequent progressive neurodegenerative disorder and affects 10 million people worldwide. As in AD, PD cases are in majority idiopathic, and protein aggregation processes represent a key pathological mechanism. The aggregation of alpha-synuclein (aSyn) as the main component of Lewy bodies and Lewy neurites may involve oxidative stress [12,13] as well as post-translational aSyn modification [14,15,16,17]. The brains of PD patients are additionally characterized by the predominant degeneration of dopaminergic neurons of the substantia nigra pars compacta, resulting in striatal dopaminergic deafferentation associated with clinical hypokinesis [18,19]. 

We contributed to the discovery of pathogenic post-translational pyroglutamate (pGlu)-modification of Aβ in AD [20] and of aSyn in PD [21], respectively. Here, we focus on the two-step enzymatic conversion of full-length aSyn into pGlu79-aSyn by matrix metalloproteinase-3 (MMP-3) and glutaminyl cyclase (QC), as schematically shown in Figure 1.

In general, MMP-3 is a zinc- and calcium-dependent endopeptidase that belongs to the stromelysin class of MMPs [22,23] and has a broad substrate spectrum for hydrolyzing components of the extracellular matrix such as fibronectin, collagen IV, proteoglycans and laminin [24]. In addition, it is implicated in pathogenic aSyn fragmentation as identified by Sung et al. [25] and Levin et al. [26] and reviewed in Bluhm et al. [27].

Under physiological conditions, QC is predominantly expressed in the hypothalamus and pituitary, where it catalyzes N-terminal pGlu modification of neuropeptides and peptide hormones such as orexin A, gastrin, gonadotropin- and thyrotropin-releasing hormones and neurotensin [28,29,30,31]. These pGlu modifications confer stability against proteolytic degradation and increase the biological activity of the respective proteins. QC was also detected in sub-populations of cortical and hippocampal interneurons [32,33,34] and in subcortical brain structures such as nucleus basalis Meynert, locus coeruleus and Edinger-Westphal nucleus [35].

Intriguingly, pathogenic post-translational protein modification by QC is implicated in AD and PD. Both the generation of pGlu-modified Aβ and of pGlu79-aSyn require the enzymatic activity of QC and promote, in both cases, protein aggregation and neurotoxicity [21,36]. Thus, this catalysis presents a shared mechanism for these neurodegenerative diseases.

In addition, there is also growing evidence for cross-disease mechanisms of pathological protein aggregation. For example, post mortem examination of brain tissue from patients suffering from a variety of neurodegenerative diseases revealed the co-existence of different misfolded proteins, raising the question of whether there might be overlaps and analogies between the specific pathologies [37,38,39,40]. Given the fact that proteins can act as homologous seeds [41,42,43], the question arose whether seeding could also take place in a heterologous manner resulting in co-aggregation of proteins characteristic for different clinical entities. In particular, the relationship between aggregating proteins typical for AD (Aβ) and PD (aSyn) has been studied [44,45,46]. In the brains of patients with AD/PD and in transgenic mice, Aβ and aSyn were shown to co-immunoprecipitate and form complexes resulting in accelerated cognitive decline [47,48,49]. In addition, aSyn immunoreactivity was reported to be associated with Aβ plaques in AD [50] and in Lewy body disease with concomitant AD pathology [51,52] as well as in pathological structures of 5xFAD [52], 3xTg [49] and Tg2576 [53] AD mouse models. 

The formation of pathogenic protein aggregates in the brain ultimately results in a glial response that may have a pro-inflammatory as well as a neuroprotective component [54,55,56,57,58]. In addition, abnormally folded proteins induce a specific protein unfolding response that includes the expression of molecular chaperones such as heat shock proteins (HSPs) [59,60]. A specific HSP27 (*aka* HSPB1) induction under conditions of Aβ and aSyn aggregate formation has been established and was shown to alleviate cellular stress and reduce neurodegeneration [61,62,63,64]. 

In the present study, we wanted to examine whether the newly identified pGlu79-aSyn variant is associated with Aβ pathology in brain tissue from AD patients and transgenic Tg2576 mice and whether this induces a heat shock protein response.

## 2. Materials and Methods

### 2.1. Antibodies

The specificities of the in-house generated rabbit antiserum against pGlu79-aSyn [21] and the goat anti-QC antiserum [65] were recently demonstrated using the respective knock-out mouse brain tissue. In addition, the commercially available antibodies against aSyn (BD Transduction Labs, Franklin Lakes, NJ, USA; Roboscreen, Leipzig, Germany; Enzo Life Sciences, Farmingdale, NY, USA), Aβ (Biolegend, San Diego, CA, USA), GFAP (Synaptic Systems, Göttingen, Germany), MMP-3 (Santa Cruz, Dallas, TX, USA; Abcam, Cambridge, UK) and HSP27 (Santa Cruz, Dallas, TX, USA) were used for immunohistochemical labelings of mouse and human brain tissue (for details see Table 1). For the analyses of mouse brain tissue, we used the mouse monoclonal Syn1 antibody, well established for aSyn staining of mouse tissue, and compared its immunohistochemical staining pattern to that of another aSyn antibody raised in a different species (K129 rabbit anti-aSyn). For aSyn immunohistochemistry in human brain tissue, the human aSyn-specific rat monoclonal antibody 15G7 was used.

### 2.2. Mouse Brain Study

#### 2.2.1. Experimental Animals

APP-transgenic Tg2576 mice developed and described earlier [66] were used as a model for amyloid pathology and to reveal a potential association of aSyn and of pGlu79-aSyn with amyloid plaques in vivo. These mice express hAPP695 with the Swedish double mutation (K670N, M671L) as transgene under control of a hamster prion protein promoter on a mixed C57BL/6 x SJL background. Mice were housed in groups of three to five animals per cage and separated by sex, with ad libitum access to water and food with 12 h day/12 h night cycles at 23 °C. The cages contained red plastic houses (Tecniplast, Buguggiate, Italy) and shredded paper flakes to allow nest building. At the age of six weeks, transgenicity of the animals was tested by polymerase chain reaction of ear punch-extracted DNA as described [66]. Mice heterozygous for the transgene (N = 6) and wild-type littermates (N = 6) were studied at the age of 18 months. All methods were carried out in accordance with the relevant guidelines and regulations.

#### 2.2.2. Tissue Preparation

Mice were sacrificed by CO_2_ inhalation and perfused transcardially with 0.9% saline, followed by perfusion with 4% paraformaldehyde in phosphate buffer (0.1 M; pH 7.4). The brains were removed from the skull and post-fixed by immersion in the same fixative for 24 h at 4 °C. After cryoprotection in 30% sucrose in 0.1 M phosphate buffer for three days, 30 µm thick coronal sections were cut on a sliding microtome and collected in phosphate buffer supplemented with 0.0025% sodium azide for storage at 4 °C.

#### 2.2.3. Immunohistochemistry

##### Single Labeling Immunohistochemistry

In order to detect Aβ, GFAP, aSyn, MMP-3, QC, pGlu79-aSyn and HSP27 in the brains of Tg2576 mice, single labeling immunohistochemistry was performed on free-floating coronal brain sections. Brain sections were washed in 0.1 M phosphate buffer (pH 7.4) for 5 min, and endogenous peroxidases were inactivated by treating brain slices with 60% methanol containing 1% H_2_O_2_ for 60 min, followed by three washing steps with 0.1 M Tris-buffered saline (TBS; pH 7.4) for 5 min each. After masking unspecific binding sites with blocking solution (5% normal donkey serum in TBS containing 0.3% Triton X-100) for 60 min, sections were incubated with the primary antibodies at 4 °C for 40 h (Table 1). When primary antibodies of mouse origin were used, the blocking solution additionally contained 3% donkey Fab fragments directed against mouse immunoglobulins to mask endogenous mouse antibodies in the processed tissue. Brain sections were then washed three times in TBS for 5 min each before being incubated with biotinylated donkey anti-mouse, anti-goat or anti-rabbit secondary antibodies, resp. (Dianova, Hamburg, Germany; 1:1000) in TBS containing 2% bovine serum albumin (BSA) for 60 min. After three washing steps in TBS for 5 min each, slices were incubated with ExtrAvidin peroxidase (Sigma, Darmstadt, Germany; 1:2000) in TBS/2% BSA, followed by washing steps and pre-incubation in Tris buffer (0.05 M, pH 7.6) for 5 min. Finally, visualization of peroxidase binding was performed by incubation with 4 mg 3,3′-diaminobenzidine (DAB) and 2.5 µL H_2_O_2_ per 5 mL Tris buffer. After washing, sections were mounted onto glass slides and cover slipped. 

##### Triple Immunofluorescent Labelings

In order to reveal the possible co-localization of full-length aSyn and of pGlu79-aSyn with the enzymes MMP-3 and QC and with Aβ, GFAP and HSP27, resp., in the mouse brain, cocktails of three primary antibodies from different species were applied as specified in Table 2. Brain sections were incubated with primary antibody cocktails at 4 °C for 40 h. Subsequently, sections were washed three times with TBS and were then incubated with cocktails of Cy2-, Cy3- or Cy5-conjugated donkey anti-mouse, -goat and -rabbit antisera (1:400 each; Dianova) in TBS containing 2% BSA for 60 min. All secondary antibodies were purchased as preparations pre-adsorbed against IgGs from multiple other species. After washing, cell nuclei were stained with Hoechst 33,342 (Sigma; 1:10,000), and sections were mounted onto glass slides before being cover slipped. In control experiments, switching the fluorescent labels of the secondary antibodies generated similar results as when following the procedure outlined above (not shown). In addition, omitting primary antibodies or replacement of primary antibodies by normal IgG from mouse, guinea pig, rabbit, rat and goat did not generate immunohistochemical labeling (not shown).

### 2.3. Human Brain Study

#### 2.3.1. Human Brain Tissue

The human brain tissue was provided by the former Brain Banking Centre Leipzig of the German Brain-Net, operated by the Paul Flechsig Institute of Brain Research, Medical Faculty, University of Leipzig. The procedure of case recruitment, the protocols, the informed consent forms and the autopsy have been approved (GZ 01GI9999-01GI0299). The diagnosis and staging of AD cases used in this study were based on the presence of neurofibrillary tangles [67] and neuritic plaques in the hippocampal formation and neocortical areas as outlined by the Consortium to establish a registry for AD (CERAD) [68] and met the criteria of the National Institute on Aging (NIA) on the likelihood of dementia [69]. For immunohistochemistry, temporal cortex (Brodmann area 22) from 6 AD cases (3 male, 3 female; all Braak stage VI) and 4 age-matched controls (2 male, 2 female; Braak stage I to III) was used. Anatomical structures and cortical layers were identified using consecutive Nissl-stained sections.

#### 2.3.2. Human Brain Tissue Preparation

Tissue blocks of the human temporal cortex were prepared in the frontal plane, fixed in 4% paraformaldehyde and cryoprotected in 30% sucrose in 0.1 M phosphate buffer (pH 7.4) before sectioning. Then, 30 µm thick sections were cut on a freezing microtome and collected in 0.1 M phosphate buffer containing 0.1% sodium azide.

#### 2.3.3. Immunohistochemistry

##### Single Labeling Immunohistochemistry

All immunohistochemical procedures were performed on free-floating brain sections. Brain sections were rinsed in PBS containing 0.05% Tween 20 (PBST), followed by pre-treatment with 1% H_2_O_2_ in 60% methanol for 60 min. Unspecific staining was blocked by treatment with PBS-T in blocking solution (PBST with 2% (*w*/*v*) BSA, 0.3% (*w*/*v*) milk powder, 0.5% (*v*/*v*) normal donkey serum) for 60 min before incubating brain sections with the primary antibodies in blocking solution at 4 °C for 40 h (for antibody concentration see Table 1). Thereafter, the tissue was transferred to a mixture of blocking solution and PBS-T buffer (1:2) containing the secondary biotinylated antibody (Dianova; 1:1000) for 60 min, followed by incubation with ExtrAvidin-conjugated peroxidase (Sigma, Germany; 1:2000) in PBS-T buffer at room temperature for 60 min. Bound peroxidase was visualized in a solution containing 2–4 mg DAB, 40 mg ammonium nickel(II)-sulfate and 5 µL H_2_O_2_ per 10 mL Tris-buffer (0.05 M; pH 8.0), yielding black epitope staining. 

##### Double Labeling Procedures

Combined immunohistochemical labeling of Aβ (DAB, brown) with aSyn or with pGlu79-aSyn, MMP-3, GFAP and HSP27 (DAB-Ni, black), respectively, was performed in a two-step labeling procedure. In the first step, single labeling of the respective Aβ plaque-associated proteins was performed following the single labeling procedure described above. After DAB-Ni staining of the respective antigens, bound peroxidase was inactivated with 1% H_2_O_2_ in PBS for 60 min to allow subsequent detection of Aβ with the mouse monoclonal 4G8 antibody, followed by a peroxidase-labeled donkey anti-mouse antibody (Dianova; 1:200; 60 min) and visualization with 8 mg DAB and 5 µL H_2_O_2_ per 10 mL Tris buffer (0.05 M; pH 7.6), resulting in a brown precipitate. The application of the DAB-Ni/DAB labeling protocol circumvents interference of autofluorescence present in human brain tissue of aged subjects with the immunofluorescent labelings [34].

### 2.4. Microscopy

#### 2.4.1. Light Microscopy

Mouse and human brain tissue sections stained immunohistochemically with DAB and DAB-Ni were examined with an Axio-Scan.Z1 slide scanner connected with a LED light source and a Hitachi HV-F202SCL camera (Carl Zeiss, Göttingen, Germany). High-resolution images from the areas of interest were taken using a 20× objective lens with a 0.8 numerical aperture (Zeiss). Images were digitized by means of ZEN 3.2 software and analyzed using the ZEN imaging tool. Photoshop CS2 (Adobe Systems, San José, CA, USA) was used to process the images. Care was taken to apply the same brightness, sharpness, color saturation and contrast adjustments in the processing of the various pictures.

#### 2.4.2. Confocal Laser Scanning Microscopy

Confocal laser scanning microscopy (LSM 880 Airyscan, Zeiss, Oberkochen, Germany) using an Axioplan2 microscope was performed to reveal the spatial relationship of pGlu79-aSyn with MMP-3, QC, aSyn, Aβ, GFAP and HSP27. For Cy2-labeled antigens (green fluorescence), an argon laser with 488 nm excitation was used, and emission from Cy2 was recorded at 510 nm applying a low-range band pass (505–550 nm). For Cy3-labeled antigens (red fluorescence), a helium–neon laser with 543 nm excitation was applied, and emission from Cy3 at 570 nm was detected applying high-range band pass (560–615 nm), and Cy5-labeled antigens (blue fluorescence) were detected using excitation at 650 nm and emission at 670 nm. Images of areas of interest were taken using a 20× objective lens with a 0.75 numerical aperture (Zeiss). Photoshop CS2 (Adobe Systems, CA, USA) was used to process the images obtained by light and confocal laser scanning microscopy. Care was taken to apply the same brightness, sharpness, color saturation and contrast adjustments in the various pictures.

## 3. Results

### 3.1. Association of pGlu79-aSyn with Aβ Plaques in Tg2576 Mouse Brain

First, using standard single-labeling immunohistochemistry, the transgenic mouse model was characterized for a potential co-localization of aSyn to Aβ plaques. In wild-type mouse brain sections devoid of Aβ plaques, resting astrocytes and a synaptic aSyn localization were observed (Figure 2). In APP-transgenic Tg2576 mouse brain sections with abundant Aβ plaque formation in the neocortex and hippocampus, however, a plaque-associated activation of astrocytes and a punctate-like plaque decoration by aSyn were detected using the Syn1 antibody (Figure 2). Additionally, the K129 antibody against aSyn showed an aSyn localization in glial cells of astrocytic morphology (Figure 2).

We next asked whether the novel pGlu79-aSyn variant and the enzymes required for its generation may also be present under conditions of Aβ plaque formation in the Tg2576 mouse brain and further induce a cellular protein unfolding response. In the wild-type mouse brain, a faint synaptic/extracellular MMP-3 immunoreactivity, QC-immunoreactive interneurons, synaptic and faint somatic neuronal pGlu79-aSyn and vascular HSP27 immunoreactivity were detected (Figure 3). Interestingly, in Tg2576 mouse brain sections, there was a diffuse accumulation of pGlu79-aSyn immunoreactivity in the periphery of Aβ plaques and in some glia-like cells (Figure 3). The enzymes implicated in pGlu79-aSyn generation, MMP-3 and QC, however, displayed different patterns of immunohistochemical labelings. MMP-3 immunoreactivity was predominantly present as distinct punctate aggregates in the periphery of Aβ plaques and rather absent from cells of glial morphology (Figure 3). In contrast, QC showed strong glial immunoreactivity in the plaque periphery as well as a diffuse extracellular appearance (Figure 3). In addition, a very strong glial HSP27 immunoreactivity was detected in association with Aβ plaques, including strongly stained glial cells (Figure 3).

In order to reveal the cellular identity of the glia-like immunoreactivity observed for pGlu79-aSyn and a potential co-localization of MMP-3, QC and HSP27 with pGlu79-aSyn, a series of triple immunofluorescent labelings were performed. Based on morphological criteria, the immunoreactivities reported above for pGlu79-aSyn, QC and HSP27 in glia-like cells were most likely attributable to astrocytes. Therefore, triple labelings also included GFAP as an astrocytic marker protein.

The triple immunofluorescent labelings in Tg2576 mouse brain sections confirmed the glial localization of pGlu79-aSyn and QC. Neither MMP-3 nor full-length aSyn were localized in glia-like cells (Figure 4a). In addition, there was an almost complete co-localization of pGlu79-aSyn with HSP27 and the majority of Aβ plaque-associated, GFAP-immunoreactive astrocytes (Figure 4b).

### 3.2. Association of pGlu79-aSyn with Aβ Plaques in AD Brain

In order to confirm the observations made in the transgenic AD mouse model in human AD brains, immunohistochemical double labelings were performed. In neocortical brain slices from both control and AD subjects, a neuronal pGlu79-aSyn immunoreactivity, in particular of layer III and V pyramidal neurons, was observed (Figure 5a). Typically, the neuronal pGlu79-aSyn labeling appeared more distinct in AD cases compared to control subjects. In addition, were observed glial and dot-like pGlu79-aSyn immunoreactivity in AD brain tissue (Figure 5a).

Neocortical control brain tissue displayed a granular MMP-3 immunoreactivity and labeling of individual QC-immunoreactive interneurons (Figure 5b). In AD, however, pGlu79-aSyn was associated with punctate and rod-like aggregates within Aβ plaques as well as in astrocytes decorating amyloid deposits (Figure 5b). Similarly, both enzymes implicated in pGlu79-aSyn formation, MMP-3 and QC, were detected in glial cells in proximity to amyloid plaques (Figure 5b).

In the Tg2576 mouse brain, HSP27 immunoreactivity was detected in cerebral blood vessels and in Aβ plaque-associated reactive astrocytes (see Figure 3). In human AD brain tissue, HSP27 was also found to be associated with microvessels and with reactive astrocytes in the proximity of Aβ plaques (Figure 6), implicating the induction of similar protein unfolding mechanisms in response to amyloid pathology. However, in contrast to observations made in the Tg2576 mouse brain, the aSyn protein did not show a granular accumulation in the amyloid plaque periphery. 

## 4. Discussion

### 4.1. aSyn and pGlu79-aSyn Aggregates in Proximity to Aβ Plaques

The present study was performed to reveal whether the pGlu79-aSyn variant recently identified in the PD brain [21] is also associated with Aβ pathology in the brains of AD subjects and its transgenic Tg2576 mouse model. Interactions between pathological proteins typical for different clinical conditions have been known for a long time and have been studied best for aSyn and Aβ peptides [46,49,70,71,72]. In particular, specific interaction sites between membrane-bound aSyn and Aβ proteins have been identified [73], and aSyn overexpression was shown to lead to the accumulation of mouse Aβ in a mouse model of PD [44]. Vice versa, Aβ was shown to promote aSyn aggregation and toxic conversion in vitro [46] and in vivo [47]. However, intracerebral injections of aSyn preparations and co-expression of aSyn in APP-transgenic mice both failed to seed but rather inhibited Aβ plaque formation [74]. Moreover, the seeding capacity of Aβ-containing brain preparations was reduced by aSyn, and Aβ deposition was ameliorated in grafted brain tissue from aSyn-transgenic mice [74].

Here, we report a distinct, punctate aggregation of full-length aSyn in the periphery of Aβ plaques in Tg2576 mouse brain but not in AD brain. The association of aSyn with Aβ plaques is well established for different APP-transgenic mouse models [49,52,53] but controversial for AD brain tissue. Although some early studies reported such an association of the aSyn non-amyloid component (NAC) domain with amyloid plaques in AD [50,75], subsequent studies using different antibodies covering other portions of the aSyn protein did not confirm an association of full-length aSyn with amyloid plaques in AD [76,77]. However, in AD cases with additional Lewy body pathology, aSyn immunoreactive dystrophic neurites decorating Aβ plaques were reported [51,52,78].

For the association of the pGlu79-aSyn variant with Aβ plaques, on the other hand, highly consistent observations were made for the transgenic mouse model and the human clinical condition. pGlu79-aSyn and the enzymes implicated in its generation, MMP-3 and QC, respectively, were consistently detected in plaque-associated structures in Tg2576 mouse and AD brain. 

### 4.2. pGlu79-aSyn, QC and HSP27 in Reactive Astrocytes

Astrocytes are generally activated under conditions of pathological protein aggregation, for example, in amyloidosis, tauopathies and synucleinopathies [79,80]. Here, in addition to the extracellular association of aSyn, MMP-3, QC and pGlu79-aSyn with Aβ plaques, immunoreactivity for QC and pGlu79-aSyn was noticed in glial cells identified as reactive astrocytes. Interestingly, full-length aSyn detected by the Syn1 antibody did not show astrocytic immunoreactivity, but the K129 antiserum generated strong aSyn labeling in reactive astrocytes. This discrepancy might be due to the different aSyn epitopes used for immunization to generate the Syn1 (aSyn aa 15–123) and K129 (aSyn aa 44–57) antibodies. Alternatively, the antibodies might recognize different aSyn cleavage products, conformations or aggregation forms such as oligomers of different sizes, protofibrils and fibrils. Indeed, oligomeric aSyn has been observed in reactive astrocytes in the A53T PD mouse model and related to blood–brain-barrier disruption [81]. Also in human PD brain, aSyn-immunoreactive astrocytes have been observed [82,83]. However, astrocytic aSyn accumulation was not detected under conditions of multiple system atrophy and cortico-basal degeneration, indicating disease and/or aSyn strain-specific aspects of astrocytic aSyn enrichment [84]. aSyn accumulation in astrocytes may subsequently contribute to their activation and aberrant function [85,86,87], as exemplified by impaired glutamate uptake [88]. In addition, astrocytic uptake of neuron-derived aSyn by endocytosis is considered a mechanism to reduce extracellular aSyn concentration and, thereby, its neuron-to-neuron propagation [89]. Finally, there are reports of aggregated aSyn transfer from astrocyte to astrocyte via tunneling nanotubes [90]. Thus, astrocytes are believed to contribute to aSyn degradation rather than spreading [91]. 

Together, our data are consistent with two different scenarios: (i) release of aSyn from neurons, its membrane-associated or extracellular truncation into Gln79-aSyn by MMP-3 and subsequent conversion into pGlu79-aSyn by QC in the extracellular space around Aβ plaques, followed by astrocytic uptake of pGlu79-aSyn (Figure 7). (ii) Alternatively, extracellular Gln79-aSyn may be taken up by local astrocytes and converted into pGlu79-aSyn by astrocytic QC (Figure 7). Aβ plaque-associated astrocytic QC expression in the brains of Tg2576 mice has been demonstrated by us earlier and was shown to be implicated in pGlu modification of thyrotropin-releasing hormone [92].

We have recently shown that pGlu79-aSyn peptides in solution instantly form oligomers that are neurotoxic [21]. Therefore, it is likely that cellular defense mechanisms become activated in astrocytes affected by pGlu79-aSyn. A typical cellular response to protect cells from cytotoxic effects of misfolded or aggregated proteins is the expression of chaperones such as HSP27. Indeed, pGlu79-aSyn in reactive astrocytes was found to be co-localized to HSP27, a member of a family of small heat shock proteins. A particular association of HSP27 with Aβ peptides has been demonstrated in cell-free assays [63,93], in cultured cells [94], in transgenic AD animal models [62] and in the brains of AD subjects [95]. HSP27/Aβ interactions also take place in reactive astrocytes. For example, primary rat astrocytes release HSP27 in response to Aβ exposure [96], and HSP27 immunoreactive cells of astrocytic morphology were observed in AD brain tissue [97]. Thus, our data on astrocytic HSP27 expression reported here for human AD and APP-transgenic mouse brain tissue are in line with a substantial amount of literature. In an APP-transgenic mouse model with amyloid pathology, one would primarily expect Aβ or pGlu-Aβ immunoreactivity in plaque-associated reactive astrocytes. Surprisingly, in double labeling experiments for Aβ or pGlu-Aβ with GFAP, we did not detect astrocytic Aβ/pGlu-Aβ localization (not shown). Instead, there was a striking astrocytic co-localization of HSP27 and pGlu79-aSyn. This observation is somewhat counterintuitive since endogenous wild-type mouse aSyn does not form pathological aggregates such as Lewy bodies and Lewy neurites. However, under conditions of Aβ pathology, the induction of astrocytic QC expression may lead to a pGlu79-aSyn generation that induces local HSP27 expression. As in the case of Aβ, there is an association between aSyn and HSP27 (for reviews, see [61,64,98]). Specifically, HSP27 was shown to bind aSyn in a specific manner [99] to protect cells from aSyn-induced cell death [100,101] and was demonstrated to co-localize with aSyn inclusions in H4 neuroglioma cells [102]. Moreover, the particularly oligomerization-prone and cytotoxic glycated aSyn variants were shown to be converted into non-toxic aSyn conformations by HSP27 [103]. This is in line with the role of HSP27 in modulating glycation-associated pathologies in synucleinopathies.

Quantitative analyses on the abundance of aSyn, pGlu79-aSyn and HSP27 in the brains of AD subjects and Tg2576 mice will be performed in future studies by biochemical methods using non-fixed brain tissue. 

## 5. Conclusions

We conclude that the recently identified pGlu79-aSyn aggregates in proximity to Aβ plaques in the brains of AD subjects and of APP-transgenic Tg2576 mice. The pGlu79-modified aSyn variant also appears to be generated in reactive astrocytes and to induce an astrocytic protein unfolding response that involves HSP27 expression. Our observations are consistent with a cross-disease mechanism of pathological protein aggregation in AD and PD. The results support the crucial role of pGlu modifications of different amyloidogenic peptides in neurodegeneration and the importance of QC for their formation. Together, this provides further evidence for the concept of QC inhibition in neurodegenerative disorders, which is currently in clinical testing. 

## Figures and Tables

**Figure 1 biomolecules-12-01006-f001:**
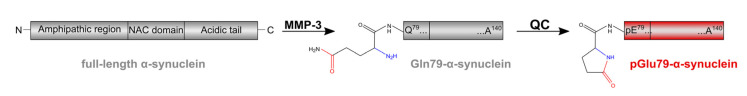
Schematic representation of sequential enzymatic pGlu79-aSyn formation. MMP-3 catalyzes proteolytic cleavage of full-length aSyn resulting in Gln79-aSyn as one fragment among several cleavage products. Gln79-aSyn then serves as precursor for QC-catalyzed pGlu79-aSyn formation (red). *Data on aSyn cleavage by MMP-3 from* [25,26]. *Data on Gln79-aSyn conversion into pGlu79-aSyn by QC from* [21].

**Figure 2 biomolecules-12-01006-f002:**
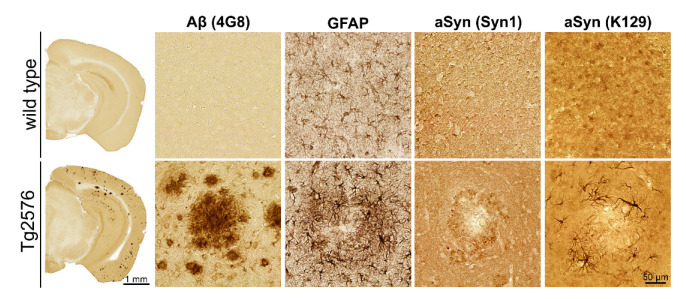
Immunohistochemical detection of Aβ, GFAP and aSyn in brains of wild-type and Tg2576 mice. In wild-type mouse brain (**top row**), no Aβ plaques, resting astrocytes and synaptic aSyn localization were observed. In brains of Tg2576 mice (**bottom row**), Aβ plaques, reactive astrocytes, punctate aSyn aggregates in the periphery of plaques (Syn1 antibody) and glial distribution of aSyn (K129 antibody) were detected. All high magnification images are from entorhinal cortex.

**Figure 3 biomolecules-12-01006-f003:**
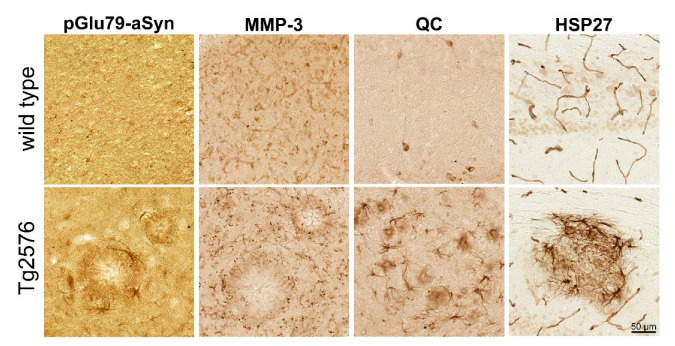
Immunohistochemical detection of pGlu79-aSyn, MMP-3, QC and HSP27 in brains of wild-type and Tg2576 mice. In wild-type mouse brain (**top row**), synaptic pGlu79-aSyn, extracellular/synaptic MMP-3, QC-positive interneurons and vascular HSP27 were detected. In contrast, in Tg2576 mouse brain (**bottom row**), pGlu79-aSyn, QC and HSP27 were detected with different intensities in glial cells. MMP-3 and in addition QC and pGlu79-aSyn appeared as punctate or diffuse extracellular aggregates in the periphery of Aβ plaques. The examples shown are from entorhinal cortex (pGlu79-aSyn and MMP-3) and from hippocampus (QC and HSP27).

**Figure 4 biomolecules-12-01006-f004:**
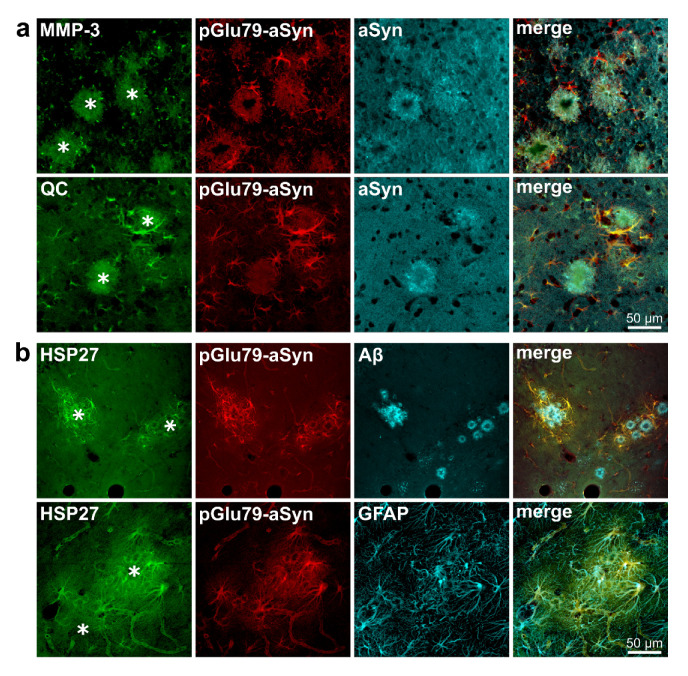
Immunohistochemical triple labeling of pGlu79-aSyn with MMP-3, QC, full-length aSyn, HSP27, Aβ and GFAP in Tg2576 mouse brain tissue. (**a**) MMP-3 and aSyn showed a punctate immunoreactivity in the plaque periphery, but no localization in cells of glial morphology. QC and pGlu79-aSyn, on the other hand, showed diffuse labeling of plaques and co-localization in glia cells. (**b**) Glial pGlu79-aSyn immunoreactivity was clearly co-localized with HSP27 and with the astrocyte marker GFAP in the plaque periphery. Asterisks indicate the position of Aβ plaques. The examples shown are from entorhinal cortex (**a**) and from hippocampus (**b**).

**Figure 5 biomolecules-12-01006-f005:**
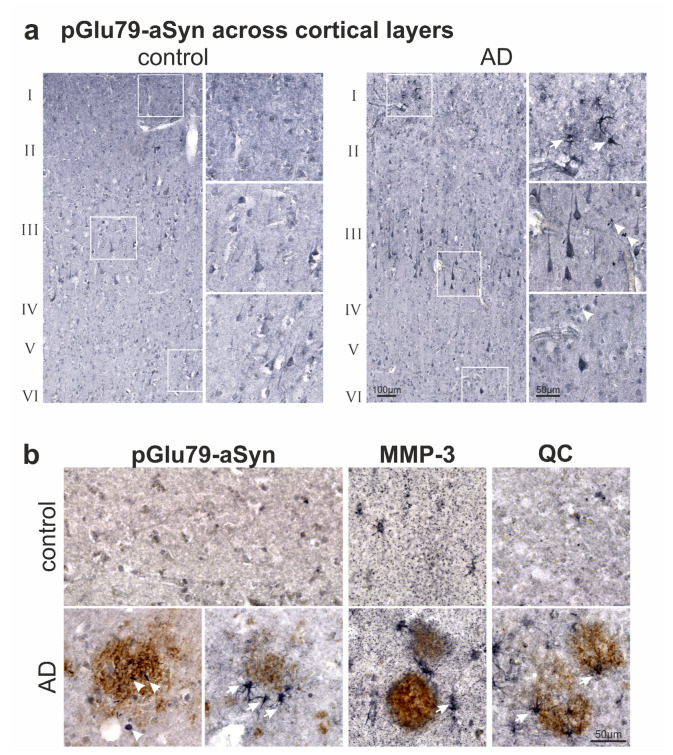
Association of pGlu79-aSyn, MMP-3 and QC with amyloid plaques in human brain tissue. (**a**) pGlu79-aSyn immunoreactivity in neocortex of a control subject (left) and of an AD case (right). The higher magnification images depict layers I, III and VI as indicated by boxes. Note the strong pGlu79-aSyn immunoreactivity in pyramidal neurons, which appears to be more distinct in AD compared to control tissue. The higher magnification images identify glial (arrows) and dot-like (arrowheads) pGlu79-aSyn immunoreactivity. (**b**) Double labelings revealed the glial presence (arrows) of pGlu79-aSyn, MMP-3 and QC (DAB-Ni, black) in proximity of Aβ plaques (DAB, brown) in AD temporal cortex (Brodmann area 22). In addition, pGlu79-aSyn immunoreactivity was found in rod-like and punctate (arrowheads) structures within Aβ plaques.

**Figure 6 biomolecules-12-01006-f006:**
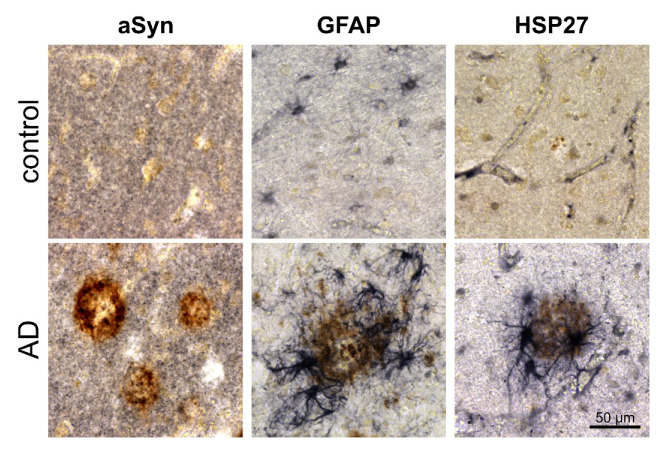
HSP27 expression by amyloid plaque-associated reactive astrocytes in human temporal cortex (Brodmann area 22). In control subjects (**top row**), synaptic aSyn labeling, GFAP-immunoreactive resting astrocytes and vascular HSP27 were detected (DAB-Ni, black). In AD (**bottom row**), strong activation of plaque-associated astrocytes and substantial astroglial HSP27 expression were evident. Full-length aSyn labeled by the 15G7 antibody, on the other hand, was not found to be enriched around Aβ plaques (DAB, brown) and was absent from reactive astrocytes.

**Figure 7 biomolecules-12-01006-f007:**
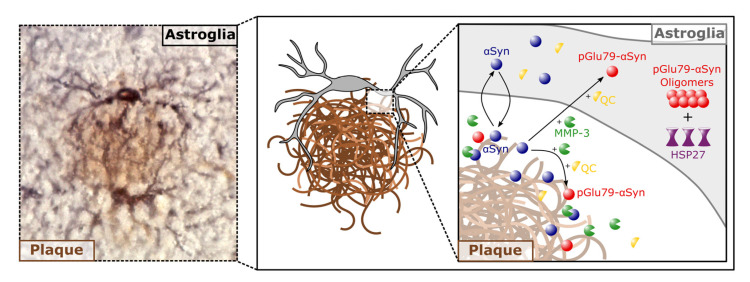
Combined immunohistochemical and schematic representation of the cellular localization of proteins investigated. (**Left**): Immunohistochemical visualization of astrocytes (GFAP, black) decorating an amyloid plaque (4G8, brown). (**Middle**): Schematic reduction of the immunohistochemical labeling. (**Right**): Zoom into the plaque, plaque periphery and astroglial interior, showing the localization of the proteins investigated in this study (aSyn, blue; pGlu79-aSyn, red; MMP-3, green; QC, yellow; HSP27, violet).

**Table 1 biomolecules-12-01006-t001:** Primary antibodies used for single labeling immunohistochemistry.

Primary Antibody (Clone/cat. #)	Company	Host	Species Reactivity	Dilution	
				Mouse	Human
Aβ (4G8/800710)	Biolegend	Mouse	Mouse, human	1:8000	1:16,000
GFAP (-/173004)	Synaptic Systems	Guinea pig	Mouse, rat, human	1:200	1:6000
aSyn; Syn1 (42/610787)	BD Transd. Labs	Mouse	Mouse, rat, human	1:8000	-
aSyn, K129 (-/0103001901)	Roboscreen	Rabbit	Mouse, human	1:200	-
aSyn (15G7/ ALX-804-258)	Enzo Life Sciences	Rat	Human	-	1:1000
MMP-3 (-/sc6839)	Santa Cruz	Goat	Mouse, rat, human	1:50	-
MMP-3 (EP1186Y/ab52915)	Abcam	Rabbit	Mouse, rat, human	-	1:3,000
QC (-/77101)	Fraunhofer IZI (Halle, Germany)	Goat	Mouse, human	1:200	1:500
pGlu79-aSyn	Fraunhofer IZI	Rabbit	Mouse, human	1:100	1:500
HSP27 (-/sc1049)	Santa Cruz	Goat	Mouse, human	1:200	1:2000

MMP-3—matrix metalloproteinase-3; QC—glutaminyl cyclase; aSyn—α-synuclein; GFAP—glial fibrillary acidic protein; HSP27—heat shock protein 27.

**Table 2 biomolecules-12-01006-t002:** Cocktails of primary and secondary antibodies used for triple immunofluorescent labeling in mouse brain tissue.

Primary Antibody	Host	Dilution	Secondary Antibody
MMP-3	Goat	1:100	Donkey anti-goat Cy2
pGlu79-aSyn	Rabbit	1:100	Donkey anti-rabbit Cy3
aSyn (Syn1)	Mouse	1:4000	Donkey anti-mouse Cy5
QC	Goat	1:100	Donkey anti-goat Cy2
pGlu79-aSyn	Rabbit	1:100	Donkey anti-rabbit Cy3
aSyn (Syn1)	Mouse	1:4000	Donkey anti-mouse Cy5
HSP27	Goat	1:100	Donkey anti-goat Cy2
pGlu79-aSyn	Rabbit	1:100	Donkey anti-rabbit Cy3
GFAP	Guinea pig	1:200	Donkey anti-guinea pig Cy5
HSP27	Goat	1:100	Donkey anti-goat Cy2
pGlu79-aSyn	Rabbit	1:100	Donkey anti-rabbit Cy3
Aβ (4G8)	Mouse	1:4000	Donkey anti-mouse Cy5

MMP-3—matrix metalloproteinase-3; QC—glutaminyl cyclase; aSyn—α-synuclein; GFAP—glial fibrillary acidic protein; HSP27—heat shock protein 27.

## Data Availability

The data presented in this study are available on request from the corresponding author.

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
