# Peer review of "Immunohistochemical Demonstration of the pGlu79 α-Synuclein Fragment in Alzheimer’s Disease and Its Tg2576 Mouse Model"

_biomolecules, 2022, doi:10.3390/biom12071006_

Round 1

Reviewer 1 Report

This is a very interesting manuscript showing the association of the pyroglutamate 79-modified α-synuclein variant (pGlu79-aSyn)  with Abeta plaques, both in the AD transgenic mouse model Tg2576 and in human AD brain  together with the enzymes implicated in its generation,  matrix metalloproteinase-3 (MMP-3) and glutaminyl cyclase (QC). This, pathogenic associated post-translational protein modification by QC is implicated in AD and previously shown for PD and presents a shared mechanism of these neurodegenerative diseases.

We would however like to see more controls in the IHC including use of an  alpha Syn antibody raised against the non modified alpha Syn peptide 79-140 and normal Ig. Also we would like to see  quantifications of the images and statistical analysis. It is not clear how many mice or human AD brain samples  were used in the images. The study is mostly correlative without any biochemical data.  Can the authors inhibit the MMP3/QC pathway and determine the presence/absence  of toxic  pGlu79-aSyn in primary neurons or astrocytes derived from the AD mouse model. 

Author Response

This is a very interesting manuscript showing the association of the pyroglutamate 79-modified α-synuclein variant (pGlu79-aSyn) with Abeta plaques, both in the AD transgenic mouse model Tg2576 and in human AD brain together with the enzymes implicated in its generation, matrix metalloproteinase-3 (MMP-3) and glutaminyl cyclase (QC). This, pathogenic associated post-translational protein modification by QC is implicated in AD and previously shown for PD and presents a shared mechanism of these neurodegenerative diseases.

Response: We thank the reviewer for the positive reception of our manuscript and for the recommendations to improve its quality.

  1. We would however like to see more controls in the IHC including use of an alpha Syn antibody raised against the non modified alpha Syn peptide 79-140 and normal Ig.

Response: This is an excellent point raised by the reviewer and of particular interest to reveal the relative abundance of the non-modified Gln79-aSyn fragment versus pGlu79-aSyn with specific antibodies. However, this is very challenging since N-terminal Gln (as needed in the immunization peptide) tends to cyclize spontaneously into the pGlu form. Control experiments (i) using normal IgGs from mouse, guinea pig, rabbit, rat and goat instead of the primary antibodies and (ii) omitting primary antibodies were performed and are now mentioned in the methods section (page 5, lines 187-189). In addition, the specificities of the aSyn and pGlu79-aSyn antibodies were validated by dot blot analyses against recombinant proteins and in brain tissue of aSyn knock-out mice (Hartlage-Rübsamen et al., 2021).

  1. Also we would like to see quantifications of the images and statistical analysis. It is not clear how many mice or human AD brain samples were used in the images.

Response: The number of mice and human subjects included in this study is given in the methods section (line134-135 for mouse brain tissue and 209-211 for human brain tissue). The images show one typical example of the immunohistochemical labellings. Quantifications of immunohistochemical labellings are frequently done in the literature, but they generally suffer from technical limitations, such as the lack of standard curves and the question of linearity between the abundance of proteins and the signals detected. Therefore, we currently plan a comprehensive biochemical follow-up study with quantifications of proteins by ELISA and/or Western blot analyses (see also point 3 and reviewer 2 point 3).

  1. The study is mostly correlative without any biochemical data. Can the authors inhibit the MMP3/QC pathway and determine the presence/absence of toxic pGlu79-aSyn in primary neurons or astrocytes derived from the AD mouse model. 

Response: We agree with the reviewer on the descriptive nature of our study, which led us select the title of the manuscript (“Immunohistochemical evidence…”). In follow-up studies, we do plan both, biochemical quantifications (see above) and manipulations of enzymatic activities (of MMP-3 and QC) using enzyme inhibitors as well as overexpression and knock-down of these enzymes. However, this goes beyond the scope of the present manuscript that focusses on immunohistochemical localization and spatial relationship of the proteins of interest.

Reviewer 2 Report

The manuscript of Bluhm et al., show the potential co-localization of pGlu79 α-synuclein fragment in Alzheimer’s disease (AD) brains and in a mouse model of AD. The results are interesting, but there are some suggestions to improve the quality of the manuscript.

1)    The colors used for the triple fluorescent staining in Fig 4 do not allow detecting co-localization of three proteins clearly, particularly using the blue/cyan fluorescent color. Perhaps it is best to show only double stainings.

2)      Does pGlu79-Syn also co-localize with AβpE3 and AβpE11?? Does pGlu79-Syn correlate with the amount of AβpE3 and AβpE11 containing plaques, so it reflects an increase of the enzymes involved in this process?

3)      It would be good to show some kind of quantification of the stainings, for instance the distance of pGlu79-Syn staining from Aβ plaques and the number of positive cells (astrocytes vs neurons) with the different antibodies for α-syn used.

4)      Is there any relationship between the stage of AD disease with the amount of pGlu-Syn deposition? Does it occur only in late stages?

5)      The statement “pGlu79-aSyn participates in β-amyloid plaque formation” is not proven in this article. It is only shown that this particular form of α-syn is present close to Aβ plaques. In fact, this is emphasized by the observation that plaque formation by overexpression of APP (in the transgenic mice) can lead to increased α-syn staining.

6)      In the discussion, it would be interesting to comment on results that show that overexpression of α-syn in APP transgenic mice resulted in reduced Aβ plaque aggregation (Bachhuber et al, Nat Med 2015 Jul;21(7):802-7).

Minor issues

1)      It would be good to add in the results section the differences between the different α-syn antibodies used and why they were chosen, which only appears in the discussion.

2)      A-beta is best if written with Greek letters (Aβ).

Author Response

The manuscript of Bluhm et al., show the potential co-localization of pGlu79 α-synuclein fragment in Alzheimer’s disease (AD) brains and in a mouse model of AD. The results are interesting, but there are some suggestions to improve the quality of the manuscript.

Response: We thank the reviewer for the positive reception of our manuscript and for the recommendations to improve its quality.

  1. The colors used for the triple fluorescent staining in Fig 4 do not allow detecting co-localization of three proteins clearly, particularly using the blue/cyan fluorescent color. Perhaps it is best to show only double stainings.

Response: We agree with the notion that the identification of individual labellings in the overlay channel is challenging (due to the possible three different co-localizations). However, since this can be identified from the single labelling channels, we would prefer to keep the Figure in its current version.

  1. Does pGlu79-Syn also co-localize with AβpE3 and AβpE11?? Does pGlu79-Syn correlate with the amount of AβpE3 and AβpE11 containing plaques, so it reflects an increase of the enzymes involved in this process?

Response: It is indeed tempting to speculate that different pE-modified peptides are enriched in parallel since their formation is QC-dependent. However, the mechanism of N-terminal truncation as another pre-requisite for pE modification differs (Abeta by aminopeptidase A, meprin-b or DPP-4 and aSyn by MMP-3). In addition, AbpE11 is barely present in Tg2576 mouse brain. We do know from a previous study that P-Ser129-aSyn co-localizes with AbpE3 (Köppen et al., 2020 PMID 32013170) but have not analyzed this for pGlu79-aSyn.

  1. It would be good to show some kind of quantification of the stainings, for instance the distance of pGlu79-Syn staining from Aβplaques and the number of positive cells (astrocytes vs neurons) with the different antibodies for α-syn used.

Response: As stated above (Reviewer 1, point 2), we currently plan a comprehensive biochemical follow-up study with quantifications of proteins by ELISA and/or Western blot analyses. However, this does not allow differentiation of signals arising from astrocytes and from neurons.

  1. Is there any relationship between the stage of AD disease with the amount of pGlu-Syn deposition? Does it occur only in late stages?

Response: All AD cases enrolled in this study were Braak stage VI, so we cannot make any statements on a possible relationship between AD stage and the degree of pGlu79-aSyn deposition. The information on the Braak stage is now given in the Methods section (page 5, lines 209-211). In future studies, however, it will definitely be of importance to analyze the temporal profile of pGlu79-aSyn deposition during disease progression.

  1. The statement “pGlu79-aSyn participates in β-amyloid plaque formation” is not proven in this article. It is only shown that this particular form of α-syn is present close to Aβplaques. In fact, this is emphasized by the observation that plaque formation by overexpression of APP (in the transgenic mice) can lead to increased α-syn staining.

Response: We agree with the reviewer that our data do not allow drawing mechanistic conclusions and have corrected this sentence (Abstract, lines 28-29).

  1. In the discussion, it would be interesting to comment on results that show that overexpression of α-syn in APP transgenic mice resulted in reduced Aβplaque aggregation (Bachhuber et al, Nat Med 2015 Jul;21(7):802-7).

Response: We now discuss the significant findings made by Bachhuber et al., 2015 (page 11, lines 391-395).

Minor issues

  1. It would be good to add in the results section the differences between the different α-syn antibodies used and why they were chosen, which only appears in the discussion.

Response: We now explain in the Methods section why we used the particular aSyn antibodies (page 3, lines 116-121).

  1. A-beta is best if written with Greek letters (Aβ).

Response: The term “Abeta” was changed to “Ab" throughout the manuscript.

Reviewer 3 Report

Bluhm et al. in their manuscript “Immunohistochemical demonstration of the pGlu79 α-synuclein fragment in Alzheimer’s disease and its Tg2576 mouse model” present novel data on aSyn pathology in AD and AD mouse model, which adds stronger evidence on PD/AD cross mechanisms. For this reviewer, the experimental design was appropriated, the interpretation of the results adequate, and the discussion well-consistent with the observations. Thus, the manuscript is recommended for publication in Biomloecules. 

This reviewer additionally has few recommendations on modifications aimed to polish and make the manuscript clearer.

Minor comments. 

-Add antibody product numbers. Describe, very shortly, the specificity (differences) of different anti-aSyn antibodies (Syn1, K129, 15gt) and why they were used earlier in the manuscript (in methods or Fig legends ?).

-For clarity, explain if the very same field, of wt/control or tg mice/AD, in Fig 1 and 2 or Fig 5b AD and Fig 6, are shown.

-Line 285: I would recommend to change “Intriguingly” to “Interestingly”. I wonder why the authors were “intrigued” as changes in the protein expression patterns were hypothesised. Otherwise clarify.

-Line 311: Sentence starting with “Neither…” please revise wording or punctuation, sentence is difficult to understand.

-Figure 4. Replace stars with other marker, with dotted line circles or arrows and depict them in all images of the series. Stars block the view of the relevant areas in some cases. Would it add more clarity if it is shown also two antibody merge of pGLu79aSyn/asyn, MMP-3/pGlu79-Syn and MMP3/aSyn separately? 

-Line 327. Language usage. Should it be more appropriated to use “confirm” rather than “validate”?

-Could (some) Abeta plaques be identified in images 5, and 6?

-Discussion. Could be added background (couple of sentences, and references) on the physiological localization of QC?

-Line 456: “However…” Could these experiments be mentioned in the result section in a more elaborate, but not long, way?

Author Response

Bluhm et al. in their manuscript “Immunohistochemical demonstration of the pGlu79 α-synuclein fragment in Alzheimer’s disease and its Tg2576 mouse model” present novel data on aSyn pathology in AD and AD mouse model, which adds stronger evidence on PD/AD cross mechanisms. For this reviewer, the experimental design was appropriated, the interpretation of the results adequate, and the discussion well-consistent with the observations. Thus, the manuscript is recommended for publication in Biomloecules. 

This reviewer additionally has few recommendations on modifications aimed to polish and make the manuscript clearer.

Response: We thank the reviewer for the positive reception of our manuscript and for the recommendations to improve its quality.

Minor comments. 

  1. Add antibody product numbers. Describe, very shortly, the specificity (differences) of different anti-aSyn antibodies (Syn1, K129, 15gt) and why they were used earlier in the manuscript (in methods or Fig legends ?).

Response: Antibody product numbers were added to Table 1. The differences of the anti-aSyn antibodies are now explained in the Methods section (page 3, lines 116-121).

  1. For clarity, explain if the very same field, of wt/control or tg mice/AD, in Fig 1 and 2 or Fig 5b AD and Fig 6, are shown.

Response: The topographical origin of the images shown in all Figures is now indicated in the respective Figure Legends.

  1. Line 285: I would recommend to change “Intriguingly” to “Interestingly”. I wonder why the authors were “intrigued” as changes in the protein expression patterns were hypothesised. Otherwise clarify.

Response: The term was changed as recommended by the reviewer.

  1. Line 311: Sentence starting with “Neither…” please revise wording or punctuation, sentence is difficult to understand.

Response: The sentence was shortened to improve readability.

  1. Figure 4. Replace stars with other marker, with dotted line circles or arrows and depict them in all images of the series. Stars block the view of the relevant areas in some cases. Would it add more clarity if it is shown also two antibody merge of pGLu79aSyn/asyn, MMP-3/pGlu79-Syn and MMP3/aSyn separately? 

Response: We understand the point raised by the reviewer and have thoroughly discussed this subject. The asterisks indicating the position of amyloid plaques indeed cover parts of them. This was the reason why we did not copy/paste the asterisk into all channels. However, we are convinced that the plaques can still be recognized easily. Importantly, the focus in Figure 4 is not on plaque morphology, but on plaque-associated glial cells.

  1. Line 327. Language usage. Should it be more appropriated to use “confirm” rather than “validate”?

Response: We think both terms reflect our intention but changed it as suggested by the reviewer.

  1. Could (some) Abeta plaques be identified in images 5, and 6?

Response: In Figure 4, we marked plaques with asterisks since they were not specifically labelled in 3 out of 4 different triple fluorescent labellings. However, in Figures 5 and 6 plaques are already labelled by the brown DAB product, as is now also indicated in the legend of Figure 6.

  1. Discussion. Could be added background (couple of sentences, and references) on the physiological localization of QC?

Response: We added some sentences and references on the physiological localization of QC in hypothalamus, pituitary, neocortex, hippocampus and subcortical brain structures. We felt this information is better given in the Introduction instead of the Discussion (page 2, lines 59-66).

  1. Line 456: “However…” Could these experiments be mentioned in the result section in a more elaborate, but not long, way?

Response: We now hope to have better explained that in a mouse model with amyloid pathology, we expected to detect astrocytic Abeta/pGlu-Abeta rather than astrocytic pGlu-aSyn peptides (page 13, lines 469-472).

Reviewer 4 Report

Your work is intresting and gives a new aspect in the common pathophysiology/molecular mechanism between AD and PD and reveal the importance of post-translational modifications of aSyn.

Of course, there are some improvments that are needed, in order to be a complete and significant work:

1) INTRODUCTION. 

-Give a quick link between astrocytes and the role in post-translational modifications of aSyn and, also, their role in AD/PD pathology.

-Give a quick introduction of MMP3 (role, localization etx),as it is an important protease that in already linked with the pathophysiology of aSyn.

-QUESTION: studies support that MMP3 can cleave the N terminus of aSyn in more that one sites (residues 54-57, 71-75 etc). Does this fact affects the reaction needed for pGlu79-aSyn creation?

2) MATERIAL & METHODS

-everything is detailed explained. Complete and solid experimental design is indicated.

3) RESULTS

-It would be really helpful for every reader, if the information that is shared from your images in this part, could be quantified and given as chat-bars or any other statistical graph type! (it can be challenging to quantify light  microscope images, but it can be done with simple (free) software. For flruorescence images it is much easier to quantify and identify colocalization etc).

-in Figure 4,  can you provide a better image for aSyn ? (high backround). A better resolution is needed.

-QUESTION: have you quantified the protein (or the mRNA) levels of MMP3, QC and HSP27 in your samples (mouse and human brain samples)? Are there any differences betwwen the Control and the patient samples?

4|) DISCUSSION

Its complete enoufh. Tries to give explanations in every major question that is rised. 

Author Response

Your work is interesting and gives a new aspect in the common pathophysiology/molecular mechanism between AD and PD and reveal the importance of post-translational modifications of aSyn.

Of course, there are some improvements that are needed, in order to be a complete and significant work.

Response: We thank the reviewer for the positive reception of our manuscript and for the recommendations to improve its quality.

INTRODUCTION. 

  1. Give a quick link between astrocytes and the role in post-translational modifications of aSyn and, also, their role in AD/PD pathology.

Response: These aspects are thoroughly outlined in the Discussion (see page 12, lines 412-434) and illustrated in the summary Figure 7.

  1. Give a quick introduction of MMP3 (role, localization etx), as it is an important protease that in already linked with the pathophysiology of aSyn.

Response: We now briefly introduce the physiological role of MMP-3 and its implication in aSyn processing in the introduction (page 2, lines 54-58).

  1. QUESTION: studies support that MMP3 can cleave the N terminus of aSyn in more that one sites (residues 54-57, 71-75 etc). Does this fact affects the reaction needed for pGlu79-aSyn creation?

Response: It is correct that aSyn can be cleaved by MMP-3 at four major cleavage sites. We have a parallel manuscript submission to another Journal on MMP-3 expression in PD mouse models where we give detailed information of aSyn fragmentation by MMP-3. Since this is not really in focus of the present manuscript, we do not expand this aspect here and would like to refer to the primary publications by Sung et al., 2005 and Levin et al., 2009 as well as to our review (Bluhm et al., 2021) (see page 2, lines 57-58).

MATERIAL & METHODS

Everything is detailed explained. Complete and solid experimental design is indicated.

RESULTS

  1. It would be really helpful for every reader, if the information that is shared from your images in this part, could be quantified and given as chat-bars or any other statistical graph type! (it can be challenging to quantify light  microscope images, but it can be done with simple (free) software. For flruorescence images it is much easier to quantify and identify colocalization etc).

Response: As stated above (see reviewer 1, point 2 and reviewer 2, point 3), quantitative analyses are planned in follow-up studies using biochemical techniques such as  ELISA and Western blots which give more reliable quantitative data.

  1. In Figure 4,  can you provide a better image for aSyn ? (high backround). A better resolution is needed.

Response: The aSyn labelling outside of plaques does not represent background but results from labelling of a myriad of synapses. We could reduce the intensity of the blue channel until only the plaque-associated labelling remains and the other parts of the image are black (as for Abeta and GFAP in part b of Figure 4), but this would then not represent the underlying synaptic labelling. Since we do not focus on synapses but on astrocytes, the resolution is sufficient.

  1. QUESTION: have you quantified the protein (or the mRNA) levels of MMP3, QC and HSP27 in your samples (mouse and human brain samples)? Are there any differences betwwen the Control and the patient samples?

Response: No, we have not but we intend to do so using unfixed brain tissue and biochemical analytical techniques.

DISCUSSION

Its complete enoufh. Tries to give explanations in every major question that is rised. 

Round 2

Reviewer 2 Report

The authors have replied to the requests raised by the reviewer. However, if they do not aim to include any type of analysis/quantification of the immunohistochemistry results, they need to add this as a potential work to do in the discussion. I still encourage the analysis, since the paper will improve.

Author Response

We now included a statement on the importance of future qunatitative analyses in the Discussion.